# Hepatitis B Virus Chronic Infection in Blood Donors from Asian and African High or Medium Prevalence Areas: Comparison According to Sex

**DOI:** 10.3390/v14040673

**Published:** 2022-03-24

**Authors:** Jean-Pierre Allain, Shirley Owusu-Ofori, Xianlin Ye, Cyrille Bisseye, Mira El Chaar, Chengyao Li

**Affiliations:** 1Department of Hematology, University of Cambridge, Cambridge CB1 8RN, UK; 2Transfusion Medicine Unit, Komfo Anokye Teaching Hospital, Kumasi P.O. Box 1934, Ghana; sowusu-ofori@kathhsp.org; 3Shenzhen Blood Centre, Shenzhen 518000, China; yexianlin90@hotmail.com; 4Department of Biology, Faculty of Sciences, University of Science and Technology of Masuku, Franceville P.O. Box 943, Gabon; cyrille.bisseye@univ-masuku.org; 5Faculty of Health Sciences, University of Balamand, P.O. Box 166378 Ashrafieh, Beirut 1100-2807, Lebanon; mira.elchaar@balamand.edu.lb; 6Department of Transfusion Medicine, School of Laboratory Medicine and Biotechnology, Southern Medical University, Guangzhou 510515, China; chengyaoli62@126.com

**Keywords:** HBV, blood donors, HBsAg, viral load, anti-HBs

## Abstract

Immune control of various infectious diseases, particularly viral, was shown to be more efficient for females than males. Response to viral vaccines (HAV, HBV) was higher in females. Data on hepatitis B virus (HBV) markers accumulated over 15 years in blood donors was stratified according to sex, including HBsAg, HBV viral load and levels of anti-HBs in areas where genotypes B and C (China), genotype D (Iran, Lebanon, Tunisia) and genotype E (Ghana, Burkina Faso, Gabon) were prevalent. HBsAg was screened by either ELISA or rapid tests, anti-HBc and anti-HBs by ELISA, HBV DNA load by a standardized method across sites. In Ghanaian children less than 5 years, HBV DNA load was significantly lower in females than in males (*p* = 0.035). In China, Ghana, Burkina Faso and Gabon blood donors, median HBsAg prevalence was ~5% and 3% in China, ~8.5% and 4.5% in Gabon, ~16% and 11% in Burkina Faso and ~11% and 7% in Ghana for male and female donors, respectively (*p* < 0.001). In HBsAg+ Ghanaian blood donors, distribution and median viral load were not significantly different between sexes; occult hepatitis B infections (OBI) were significantly more frequent in males. In Chinese blood donor anti-HBc+ and anti-HBs+, anti-HBs levels tended to be higher in males but vaccinated donors’ anti-HBs+ only, while anti-HBs levels were females > males. In areas where genotypes B-E are dominant, the prevalence of chronic HBV infection (HBsAg+) seems better controlled before age 16–18 by females infected vertically or horizontally. OBIs appear considerably more frequent in men, suggesting lower efficacy of HBV infection control. Female blood donors appear significantly safer from HBV than males, and their donation should be encouraged.

## 1. Introduction

Many reports indicate that female control of infectious diseases, particularly viral, or response to viral vaccines, was better and more effective than that of males [1,2,3]. Such gender difference has been attributed to a range of factors, including genetics, hormones and cytokines [4]. Studies of Hepatitis B virus infection have long identified that females develop immune control more efficiently than males, exemplified by lower HBsAg prevalence, sooner and more frequent seroconversion to anti-HBe and distribution of viral load shifted toward lower values [5,6,7,8]. In addition, several studies have revealed that, as other anti-viral vaccines, response to HBV vaccination was higher in females than males [9,10]. However, most such studies focused on explaining the well-known very high male/female ratio of hepatocellular carcinoma [11]. Little is known about gender differences in asymptomatic individuals, such as blood donors. Blood donations being screened for HBV markers and data analyses according to sex might generate useful information.

In order to address this issue, we retrospectively examined HBV marker data collected over the last 15 years in blood donor populations from high or medium prevalence areas, such as West Africa, the Middle East and China, covering infections with genotypes B, C, D and E. In each country involved, blood donors are only (China, Burkina Faso, Gabon, Iran) or predominantly (Lebanon and Ghana) volunteer non-remunerated or family donors. Prior to donation, candidate donors having had known viral hepatitis or jaundice are excluded. Depending on available data, viral load (VL) distribution, HBsAg prevalence and level of anti-HBs were stratified according to donor sex.

## 2. Materials and Methods

### 2.1. Sample Collection

Samples included in the study were collected either as serum, such as pre-donation screening in Ghana or post-donation screening in Burkina Faso or Gabon or plasma collected after centrifugation of EDTA whole blood in other countries. HBsAg screening was performed on fresh samples in the context of routine blood donor testing or for HBV DNA or other assays on frozen plasma samples kept at −20 °C or below for no more than 1 year. Data was collected in the context of multiple previously reported studies [12,13,14,15,16,17,18,19,20,21,22]. Data was analyzed after stratification according to sex and age. Plasma samples from Ghanaian children were collected in the context of the BOTIA (Blood and Organ Transmitted Infectious Agents) studies pre-transfusion as previously described [23].

### 2.2. HBsAg Testing

HBsAg was screened in blood donors either with rapid tests as the only detection method in Ghana and as pre-donation screening in China or with enzyme immunoassays (EIA) in Burkina Faso, Gabon, Iran, Lebanon, Tunisia and China as previously described [12,13,14,15,16,17,18,19,20,21,22].

### 2.3. HBV DNA Testing

HBV DNA was detected and quantified according to previously described methods based on real-time PCR (QPCR) with a limit of detection (LOD) of 10–20 IU/mL [18]. Genotypes were identified by amplification and sequencing of at least 500 nucleotides in the *S* region, 1000 nucleotides in the *pre-S/S* region or full genome sequences [12,18].

### 2.4. Anti-HBs Quantification

Anti-HBs was quantified as previously described in Chinese blood donors aged 18–25 at the Shenzhen blood center and the Guangzhou blood center [12,24]. In this study, donors aged 18–21 were born after 1992 and were assumed to have been vaccinated against HBV. Older donors being born prior to mandatory vaccination were assumed naturally infected.

### 2.5. Ethical Approval

All studies from which data was extracted and analyzed in the present study had been approved for HBV scientific or routine examination by the local respective ethics committees as reported in each individual study. Each participating institution obtained written or verbal donor approval to utilize test results for research.

### 2.6. Study Design

The present study examines the assumption that, in random blood donors or in unselected children, prevalence or levels of HBV markers, such as HBV DNA in HBsAg positive individuals, HBsAg, or anti-HBs in HBV naturally infected or vaccinated adults differed between male and female participants.

Samples originated from areas with different dominant HBV genotypes such as genotypes B and C in China, genotype D in Iran, Lebanon and Tunisia, genotype E in West (Burkina Faso and Ghana) and central Africa (Gabon). Examining sex and occult HBV infection (OBI), data from Europe and South Africa were included [25,26,27]. In addition to potential differences in HBV markers according to sex, data from several studies was also stratified by age.

### 2.7. Statistical Analysis

Computer software (SPSS, Version 21.0, SPSS, Inc., Chicago, IL, USA) was used for statistical analysis. The data obtained was analyzed by chi-squared test and *t* test. Abnormal distribution data was analyzed by multiple-sample nonparametric test. A *p* value of less than 0.05 was considered significant.

## 3. Results

### 3.1. HBV Viral Load in Children below 5 Years of Age

In a small population of 12 female and 16 male young children, the median viral load (VL) was 17,300 IU/mL in females and 23,300,000 IU/mL in males (Figure 1). The difference was significant (*p* = 0.035).

### 3.2. Viral Load according to Sex in Adult Blood Donors

VL data was available from previously published studies in Shenzhen, China, where HBV genotypes B and C are prevalent [12], Kumasi, Ghana, where genotype E is prevalent [18,19], and Tehran, Iran and Tunis, Tunisia, where genotype D is dominant, as previously described [16,21]. In each separate study, VL was quantified with the same, previously described, QPCR assay [18]. In Kumasi Ghana, out of 189 male and 51 female samples, the percentage of female with VL below 100 IU/mL was 44.2% and 30% in males, but the overall VL distribution was not significantly different between sexes (Figure 2).

The median VL was 101 IU/mL for females and 284 IU/mL for males. The difference was not significant (*p* = 0.16). In 350 samples from Iran and Tunisia (301 males and 36 females, genotype D), the median viral load was 102 IU/mL for males and 221 IU/mL for females (*p* = 0.057) (Figure 2). In South China, VL of 163 HBsAg positive blood donors was quantified (rapid test reactive samples were excluded as not available). The median viral load of 47 females was 52.4 IU/mL and was 95.3 IU/mL for 116 male blood donors. The difference was not significant (*p* = 0.65) (Figure 2).

### 3.3. Sex Distribution in Blood Donors with Occult HBV Infection (OBI)

OBI is defined as individuals negative for HBsAg but carrying low levels of HBV DNA, excluding window period infections. We have conducted several studies to detect and characterize OBIs in several regions of the world where genotypes A to E were dominant. Results are shown in Table 1.

Irrespective of HBV genotype, there is a consistently higher percentage of males with OBI. The compiled prevalence was 81 of 1,143,201 males and 19 of 537,438 females (*p* = 0.005). Across studies, the male to female ratio ranges between 2 and 12. Such ratios are independent of sex distribution of blood donors, which ranged between 0.6 in Hong Kong and 3.2 in Ghana (Table 1) at the time studies were conducted.

### 3.4. Distribution of HBsAg Prevalence According to Age and Sex of Random Blood Donors

Blood donors were systematically screened for HBsAg with a range of assays varying in sensitivity and specificity. In sub-Saharan Africa, rapid tests are often used in small blood centers, microplate ELISA or automated EIA in larger blood centers. In China, HBV screening is done in two steps: first a rapid HBsAg test selecting negative individuals allowed to give blood that is subsequently re-tested with a higher performance EIA [12].

Table 2 and Figure 3 show results of HBsAg screening in two blood centers in Burkina Faso (Bobo-Dioulasso and Fada-Ngourma), the main blood center in Libreville, Gabon, a medium size regional hospital in Kumasi, Ghana and the Shenzhen blood center in the Guangdong province of South China. Testing results are stratified by sex and age. In each location, females showed significantly lower prevalence of the marker (*p* < 0.01). Except in Burkina Faso, HBsAg prevalence tended to increase with age irrespective of sex and to decline with donors above 40 or 50 years of age, although the relatively small number of donors in these age groups, in particular females, makes this trend unsupported by statistical analysis.

In Ghana, in 93 samples from young donors (16–19 years), the prevalence of anti-HBc was 59.1% in males (*n* = 66) and 59.3% in females (*n* = 27). In a population of 873 Chinese donors aged 18–25, prevalence of anti-HBc was 22.1% (*n* = 484) in males and 26.7% in 389 females. This data suggests that young blood donors in both Ghana and China carry similar prevalence of anti-HBc but significantly lower HBsAg prevalence in females.

In Burkina Faso, first time donors were 89% in 2009. In Gabon, 76.4% were males, and 36.6% were volunteer non-remunerated donors (VNRD) in 2009–2016.

In Ghana, only family/replacement donors (FRD) and VNRD populations with less than 20% repeat donors between 2016 and 2018 were computed.

### 3.5. Humoral Immune Response to Natural HBV Infection or HBV Vaccination at Birth

Quantification of anti-HBs was performed in Shenzhen and Guangzhou blood centers, China, on blood donor plasmas anti-HBc-positive (presumably naturally infected with HBV) and donors born after 1992 (year of mandatory HBV vaccination at birth). When aggregating 1612 donors aged 18–25 (774 females and 838 males) carrying both anti-HBc and anti-HBs, the median level of anti-HBs was 460 IU/L for females and 495 IU/L for males. In 355 donors (156 females and 199 males) carrying only anti-HBs (presumably HBV vaccinated), the median level of anti-HBs was 238 IU/L for females and 161 IU/L for males. The difference in anti-HB titers between anti-HBc positive and negative samples was highly significant (*p* < 0.001), being higher in anti-HBc positive samples (median titer anti-HBc negative = 110.2 IU/L and anti-HBc positive = 242 IU/L). In contrast, among anti-HBc negative samples, anti-HBs levels were not significant different between females and males (*p* = 0.77) but higher in females. Among anti-HBc positive samples, anti-HB levels were not significantly different between sexes (*p* = 0.155), but levels were higher in males. The median levels of anti-HBs stratified by age in presumably vaccinated (18–21 years) or not-vaccinated (22–25 years) male and female blood donors did not show any pattern suggesting higher female anti-HBs levels (Figure 4).

In the younger, presumably vaccinated, donor group (18–21 years), the male/female ratio in anti-HBs-only donors and donors negative for both anti-HBc and anti-HBs were similar (1.15 and 1.33, respectively), suggesting that not only females did not carry significantly higher levels of anti-HBs but also did not carry detectable anti-HBs for longer than males.

## 4. Discussion

The superiority of female over male immune system has been demonstrated in many situations over several decades [1,2,3,4,5,6]. It was clearly shown for response to viral vaccines, including HBV [1,2] and attributed to a number of factors, such as androgen and estrogen hormones and receptors impacting on transcription and viral replication [6]. Such mechanisms explained the long-known difference in prevalence of hepato-cellular carcinoma (HCC) between sexes [8]. In West Africa, where the prevalence of HBsAg is the highest in the world, HCC is the most frequent cancer in males but far behind cervical and breast cancer in females [28]. The HCC male to female ratio is 9:1, and viral load appears to be an important independent risk factor but not a cause on its own [8].

In terms of markers of HBV in children, several studies examined levels of response to HBV vaccination according to gender and found a significant advantage for females [29]. Viral load in pediatric chronic HBV infection was not specifically quantified. Anti-HBs levels post-vaccination were also studied in adults and higher response was observed in females [30]. Anti-HBs response kinetics over time in both children vaccinated at birth or at later ages showed progressive decline with undetectable levels beyond 5 years but this was not stratified by sex. However, there is no reason why IgG level decline should differ between sexes.

The re-examination of a series of HBV epidemiologic studies conducted by the Cambridge virology laboratory in collaboration with multiple groups in Africa and Asia provided a new set of data examining HBV infection markers according to sex. These apparently disparate studies had in common that viral load quantification was performed in a single laboratory or, as in China, with the same method transferred locally. HBsAg detection with rapid tests as well as with manual or automated EIA provided highly specific results, although differences in sensitivity have been clearly shown [20]. The data generated therefore appears comparable irrespective of the geographic origin of the samples at the date of study. It is a limitation of the study that in none of the countries where data was collected, the number of screened samples and the prevalence of HBsAg did not allow to reach statistical significance. However, the trends observed were highly reproducible between countries and HBV genotypes.

Despite the small number of subjects, young HBsAg-positive Ghanaian children carried a significantly lower VL data assembled here in females than in males (Figure 1). There are a few reports examining HBV VL in infected young children, but none were stratified according to sex [31]. The present data seems to be the first piece of evidence suggesting a higher level of early control of HBV genotype E infection after vertical or horizontal transmission in females less than five years old.

Short of published data examining the follow-up of early HBV infection in older children, the VL data here assembled examining blood donors 16 years or older indicates few differences between sexes in populations predominantly infected with genotypes B, C, D or E (Figure 2). However, in donors infected with HBV genotype D, borderline lower VL was observed in females. The paucity of female donors in both Lebanon and Iran and to a lesser extent in Tunisia did not allow further stratification according to age. A large study including over 4000 individuals conducted in Taiwan where HBV genotype B is dominant, found a highly significant lower VL in females than males carrying chronic HBV infection [8]. In the relatively small cohorts presented here, this correlation was not clearly observed and may explain non-significant difference of results.

OBI is considered an HBV infection in which viral DNA, but not HBsAg, are in circulation in chronically infected individuals. To a large extent, this condition reflects partial control of HBV replication unless HBsAg remains undetected for lack of release from hepatocytes [32] or lack of HBsAg detection related to undetectable levels or unusual amino acid substitutions [33]. Irrespective of the origins involved, the significant dominance of male OBI averaging around 80% further support more effective female control of the infection (Table 1).

In contrast, a massive difference in prevalence of HBsAg according to sex was found in all countries studied, whether genotype B/C, D or E was prevalent (Table 2, Figure 3). The variability in performance of the wide range of assays utilized does not affect noted differences according to sex although rapid tests being less sensitive than ELISAs has bearing on the overall prevalences reported. Except in China where donors less than 20 years presented similar HBsAg prevalence irrespective of sex, the significant differences observed were already established by age 16–18 in three blood donor populations tested in Central and West Africa where genotype E is dominant (Figure 3). In African countries where both vertical and horizontal transmissions during childhood are present, lower HBsAg prevalence already clearly established by age 16–19 suggests that recovery from infection indicated by HBsAg disappearance occurs in females early in life and is maintained lifelong. This is further supported by the similar high prevalence of anti-HBc in young donors irrespective of sex (59%). Similar high prevalence in donors below age 20 was previously reported (65% in 2002) [20]. Despite the systematic HBV vaccination program in place in China since 1992, the prevalence of anti-HBc remains high and similar between male and female young donors [12]. In China, Ghana and Gabon, HBsAg prevalence tends to increase with age in both sexes, with a peak around age 40, possibly reflecting added chronic infections related to sexual activity past childhood. This trend was not observed in Burkina Faso, to some extent explained by the small number of female donors beyond age 40. The general decline of HBsAg prevalence beyond age 40 has been previously reported [34]. The lower HBsAg prevalence in females and lack of prevalence decline with age had previously been described in Southern Chinese adults [35]. The data presented as well as previous reports suggest that, early in life, female exposure to HBV is similar to males, as exemplified by high and similar prevalence of anti-HBc, but the former tend to better control HBV infection than males, taking both VL and HBsAg as markers, the first marker at lower levels and the second becoming undetectable, indicating immunological control of the infection.

The last marker potentially involved in HBV infection control examined in this study was the presence and level of anti-HBs. This marker was investigated only in relatively small cohorts of Chinese blood donors. The fluctuations of median antibody level between 18 and 25 do not seem to be affected in the presumably vaccinated cohort less than 22 years of age (Figure 3). Sex does not seem to make much difference either. This limited amount of data does not match previously reported studies on female immune response to viral vaccines, including HBV [29,36].

## 5. Conclusions

In terms of blood safety, the data presented here clearly shows that female donors are somewhat safer than males, and female donation should be encouraged on that basis. This safer female trait is not limited to HBV but also applies to HCV as it was shown that the percentage of females spontaneously recovering from HCV infection (confirmed anti-HCV with undetectable viral RNA) was significantly higher than in the general population of chronically infected adults [37]. The data presented emphasizes the apparent higher efficacy of the female immune system against HBV and probably other viral infections.

## Figures and Tables

**Figure 1 viruses-14-00673-f001:**
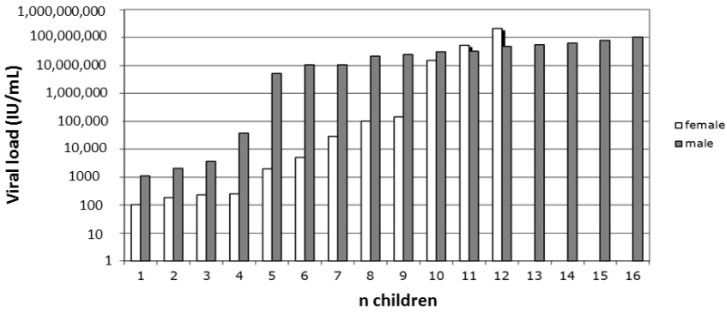
Distribution of HBV viral load in 12 female and 16 male Ghanaian children <5 years. Grey bars indicate male children and open bars female children.

**Figure 2 viruses-14-00673-f002:**
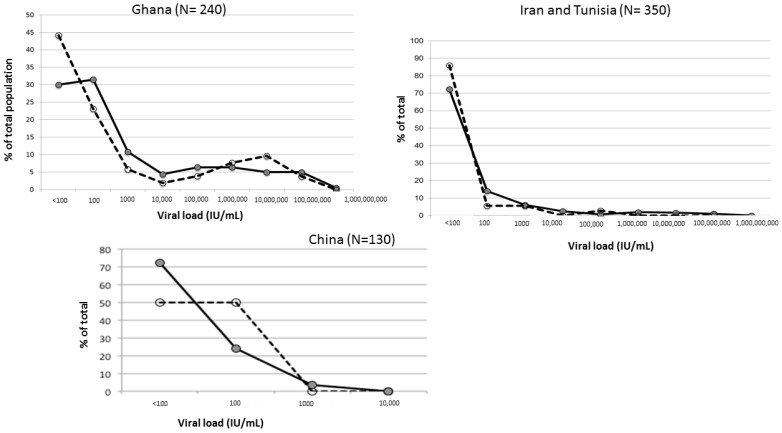
Distribution of viral load between male and female blood donors in Kumasi, Ghana (genotype E), Iran and Tunisia (genotype D), and China (genotype B and C). In all 3 graphs, grey circles and solid line indicates males, and open circles and dotted line indicate female blood donors. Viral loads are grouped as <100 IU/mL, 100–999 IU/mL, 1000–9999 IU/mL, 10^4^–99999 IU/mL, 10^5^–0.99 M, 10^6^–9.9 M IU/mL, 10^7^–99.9 M IU/mL and 10^8^–999 M IU/mL.

**Figure 3 viruses-14-00673-f003:**
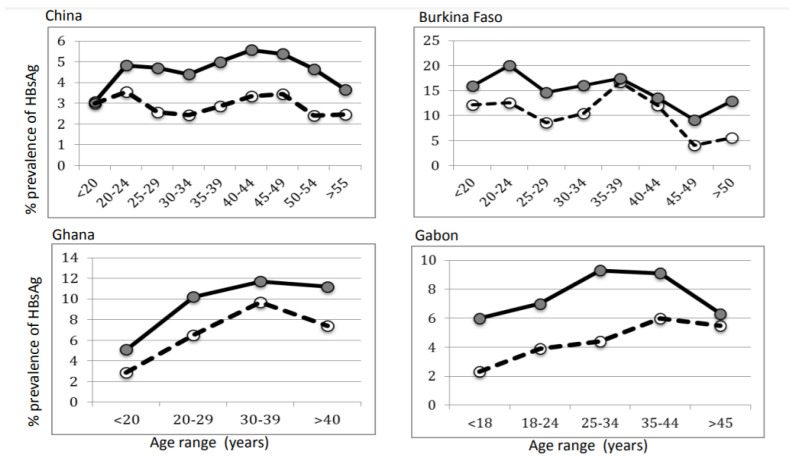
Distribution of HBsAg prevalence in blood donors stratified by age. Data are collected from 244,275 donors in Shenzhen, China; 12,032 in Burkina Faso; 75,864 in Gabon; and 14,416 donor samples in Ghana. Grey circles and solid line indicate male donors; open circles and dotted lines indicate female blood donors.

**Figure 4 viruses-14-00673-f004:**
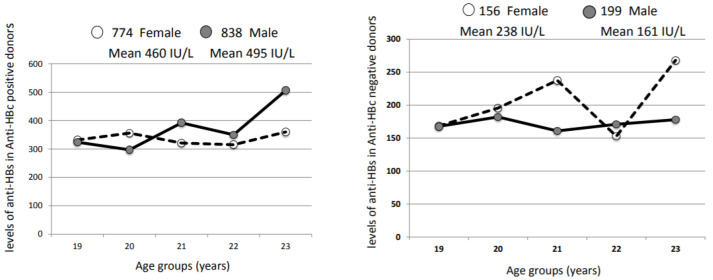
Distribution of median anti-HBs levels in presumably vaccinated Chinese blood donors 18–21 years old and non-vaccinated (22–25 years) associated or not with anti-HBc. Open circles and dotted line indicate female data; grey circles and solid line indicate male data. The numbers above graphs indicate donors in each age group for males and females.

**Table 1 viruses-14-00673-t001:** Gender distribution in blood donor samples carrying OBI from different countries.

Country	Reference	Dominant Genotype (s)	Male	Female	M/F OBI Ratio	M/F Donor Ratio	OBI Frequency
China	12	B/C	28	6	4.7	1.7	1/7517
Hong Kong	25	B	56	21	2.7	0.6	1/4255
Taiwan	25	B	16	8	2	1.8	1/1280
Thailand	25	C	16	6	2.7	3.0	1/12,807
Italy	26	D	12	0	12	2.5	1/10,450
Poland	26	D/A2	19	2	9.5	2.9	1/14,717
Spain	26	A2/D	15	2	7.5	1.2	1/6669
Ghana	20	E	28	6	4.7	3.2	1/78
RSA ^1^	27	A1	42	12	3.5	1.5	1/21,287

^1^ RSA = Republic of South Africa.

**Table 2 viruses-14-00673-t002:** Distribution of donors according to sex and type of donors.

Country	*n* Donors	*n* Male	% HBsAg+	*n* Female	% HBsAg+
Burkina Faso	12,032	8854	17.0	3178	11.8
Gabon	75,864	58,637	8.28	17,227	4.46
Ghana	14,416	10,831	9.63	3585	4.74
China	244,275	147,507	4.67	96,768	3.00

## Data Availability

Not applicable.

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
