# Peer review of "Hepatitis B Virus Chronic Infection in Blood Donors from Asian and African High or Medium Prevalence Areas: Comparison According to Sex"

_viruses, 2022, doi:10.3390/v14040673_

Round 1

Reviewer 1 Report

Please see attached

Author Response

Answers to Reviewer 1

Intro: I think it would be important to include context to blood donation in each of these countries—ie, are donors offered money? Are donations required if a loved one is ill? Are they voluntary? This could have implications for the pool from which you’re drawing conclusions – ie, people who know they’re HBV positive may not donate; people who have to donate in the context of a loved one requiring blood will give regardless.

Reply. These issues have been included: In each country involved, blood donors are only (China, Burkina Faso, Gabon, Iran) or predominantly (Lebanon and Ghana) volunteer non-remunerated or family donors. Prior to donation, candidate donors having had known viral hepatitis or jaundice are excluded.

Given your whole premise is that female immune systems are different, you should probably give a couple of examples of this directly (not just with a group of references). And highlight some of the possible mechanisms. Then when you go into the observations around HBV and the M:F ratios of HCC, it’s got more of a basis.

Reply. We believe that expending further on this is more appropriate in the discussion.

Line 41 – needs another sentence for transition – there’s a bit of a jump there.

Reply. Blood donations being screened for HBV markers, data analyses according to sex might generate useful information

I know each of the prior studies data would have had ethics approvals, but would be good to at least comment that their respective institutions had addressed that.

Reply. Each participating institution obtained written or verbal donor approval to utilize test results for research.

Figure 3 would actually be better represented as a scatter plot, comparing M and F with bars for their respective average.

Reply. A scatter plot was initially used but data appeared too busy for easy grasp of the data.

Line 173 on – this comparison of anti-HBs among HBcAb + and - is an odd one. You’re essentially comparing those who developed immunity through natural infection and those via vaccination (which to help the reader may be useful to state). This may be a point for discussion then to highlight why this may be (lots of reasons why this may be so for HBV – interesting actually that it’s not higher – think integrated DNA and HBsAg production even if cleared peripheral viral loads). This anti-HBs with anti-HBc must clearly be a subset of those individuals but also likely represent a different population –you’re referring to people that had likely been infected as a neonate, taking decades to lose HBsAg, then finally seroconvert to produce anti-HBs. Still, interesting to see the M/F breakdown, but you can’t really compare the core+ and – .

Reply. The interpretation of this data appears in the discussion. As suggested by the reviewer, it suggests that immune response to natural infection is similar between sexes while response to vaccination is significantly higher in females, as reported in the literature. Admittedly, natural infections are seen in an older population than those vaccinated but this difference should not affect response according to sex.

Table 1 – why do you have a footnote in your table highlighting that there is no M/F OBI ratio? Perhaps this is mislabelled? You claim a difference in your discussion.

Reply. We agree with the reviewer that this footnote is not necessary and it was removed.

Line 199 – take it step further—if you’re saying this is the link, then provide context with your results – If the link is viral load and females have lower VLs, then this could support the direct link. Alternatively, it’s not a viral load thing and it’s actually the female’s immune system. Either or both could work and these are some of the conclusions of your data it seems.

Reply. We do not believe that lower viral load in females is explanatory on its own of the sex prevalence difference in HCC; other factors involving the whole immune system is probably at play but there is not specific reports to support this hypothesis.

Line 203 – the sentence is kind of free-floating. Either elaborate or remove.

Reply. The sentence has been rephrased: The HCC male to female ratio is 9:1 and viral load appears an important independent risk factor but not a cause on its own [8].

Re: anti-HBs, An important factor that’s not brought up here is the idea of antibody contraction. It is a normal process for antibody titres to decrease over time – it doesn’t mean that someone is “losing immunity” as is unfortunately the interpretation (and a well-over interpreted situation of the general and scientific public from all our SARS-CoV2 vaccination studies, etc). This is an important step for our immune system to conserve resources and “contract response” or decrease output of HB-specific Abs. It doesn’t mean the B cells that make these Ab’s have been lost. This therefore makes it hard to compare someone who’d just received a vaccination to someone who’d been vaccinated as a child (you can try to compare initial responses, yes, but when decades have past, those lines blur and I would argue, not comparable). It also makes it hard to make any conclusions around better or worse immunity based on these levels. So I would strongly caution as to what conclusions are being made around this marker specifically. More Ab does not = more immunity, full stop.

Reply. This issue has been added to the discussion as follows: Anti-HBs response kinetics over time in both children vaccinated at birth or at later ages showed progressive decline with undetectable levels beyond 5 years but this was not stratified by sex. However, there is no reason why IgG level decline should differ between sexes.

Limitations – these are partly described--could be their own section. -nature of the blood donations; available numbers from which to draw conclusions;

Reply. The following sentences were added: It is a limitation of the study that in none of the countries where data was collected, the number of screened samples and the prevalence of HBsAg did not allow to reach statistical significance. However, the trends observed were highly reproducible between countries and HBV genotypes.

Line 222- “VL data assembled here”

Reply. Sentence was modified: Despite the small number of subjects, young HBsAg positive Ghanaian children carried a significantly lower VL data assembled here in females than in males.

Line 241- I haven’t heard of OBI being described as a “multi-origin” HBV infection before. What does that even mean? Would revisit your definition.

Reply. This qualifier has been removed.

Line 244- wording – this would again be fixed by re-wording your definition. A situation where HBsAg is undetected because of mutation is not OBI, but rather a misinterpretation, and to this end, how common is such an occurrence?

Reply. The definition of OBI includes a negative HBsAg detection. Such negative result might originate from undetectable level of the marker or variants escaping detection with the assays used. The former has been added to the sentence: related to undetectable levels or unusual amino acid substitutions.

Line 245 – “mechanism of OBI” is not the correct term for this in the sense you’re using it.

Reply. The term has been replaced by ‘origin’.

Line 246 - OBI M/F ratio – your Table 1 says there’s no correlation – need to clarify

Reply. The reference to OBI correlation has been removed from Table 1.

Line 248/249 – “strongly suggests recovery from infection in females occurs early ...and maintained lifelong”. This statement seems a little bold from the data supplied to support it – and I’m not sure “recovery” is the correct term either. Perhaps seroconversion or HBsAg loss if you can claim that’s what you’re looking at. Might be better to talk about trends if you don’t have the data.

Reply. The word ‘strongly’ has been deleted. Recovery from infection has been qualified by adding: indicated by HBsAg disappearance.

Line 253/254 – “In China, Ghana and Gabon, HBsAg prevalence tends to increase with age in both sexes with a peak around age 40, likely reflecting added chronic infections related to sexual activity.” WHAT?! Or how about that the older donors just did not have access to vaccination or perinatal protection like the younger. Are you claiming that people specifically from those countries are more sexually active and are all therefore acquiring new STIs as this marker? There is much data in the STI literature to contradict this statement – please re-word. Also, your following line about seeing less HBsAg pos donors after age 40 – again, consider why this may be –if a patient learned they were positive, through, say blood donation, perhaps they wouldn’t go back to donate; perhaps comorbidities accumulated after age 40 may reduce the chance that someone would donate (and esp if HBV pos with potential morbidity associated with that) and if you take into account the natural history of HBV – yes, you also see loss of HBsAg over time...and for those infected from birth (the most common route of HBV transmission by far), you may start to see HBsAg loss around the 3rd or 4th decades...

Reply. The sentence has been modified by replacing likely by ‘possibly’ and adding ‘past childhood’ at the end.

Line 262/263 – I agree these are likely the most interesting comparators. Sentence is a little awkward, though – perhaps re-word.

Reply. The sentence was kept without changes.

Conclusion:
“In terms of blood safety, the data presented here clearly shows that female donors are safer than male and female donation should be encouraged on that basis.” This is a bit of a jump! And is NOT the conclusion you should be drawing from this. Lower HBV viral load in a blood donation is still infectious! Just because it came from a female donor, doesn’t make it any safer! I thought I was going to be reading conclusions about female immunity in the context of HBV from the populations under study, not an all-encompassing statement about preferred blood. All blood donations are precious and can be life-saving. If properly screened they can also lower the risk of longer- term complications and secondary infections.

Reply. The statement made has to do with the fact that the data presented are from blood donors and that, in developing countries, the proportion of female donors is generally low, contrary to most developed countries where female donors are the majority. A sentence emphasising the higher immune efficacy of females against viral infections was added: The data presented emphasizes the apparent higher efficacy of the female immune system against HBV and probably other viral infections.

Reviewer 2 Report

In this study by Allain et al, they compared the prevalence of markers for hepatitis B between men and women in areas of high prevalence of HBV infection. Overall, this is an interesting manuscript and describes important aspects of HBV epidemiology. However, I have some important comments and several small ones to improve the quality of the manuscript for publication.

Title
The authors should be more specific in citing the location (continent) of areas of high HBV prevalence, as they were only China, Iran, Lebanon, Tunisia, Ghana, Burkina Faso and Gabon.

Abstract
Abstract is confused. The study's objectives are unclear. Authors should decrease the amount of results (such as p-values, for example). It has a lot of information that fits better in the introduction.
The title does not match the information in the abstract. The authors describe many serological markers, such as anti-HBs+, that are unrelated to chronic hepatitis B. How important is information regarding Ghanaian children less than 5 years in the context of blood donors? What is the importance for the manuscript of information regarding the vaccinated in the context of chronic infection? What is the methodology used in the study? Include statistical analysis.

Introduction
The introduction is very brief. The authors better situate the reader in relation to the choice of countries that are part of the study. For example, what determines that an area has a high, medium, or low prevalence of HBV infection? I emphasize here that in the abstract and in the title, the authors mention that they will carry out the study only in areas of high prevalence and in the objectives at the end of the introduction they also refer to areas of medium prevalence. Other points that could be included in the introduction concern: occult hepatitis B, meaning of serological markers, definition of chronic HBV infection and information regarding blood donors and HBV genotypes.

Line 38: the authors place several studies and present only one reference. You should have more.

The objectives are still not very clear. Will the study period be 15 (abstract) or 20 years (introduction)?

In Material and Methods

Line 49: The sentence "Samples included in the study were collected either as serum such as pre-donation screening in Ghana or post-donation screening in Burkina Faso or Gabon or plasma collected after centrifugation of EDTA whole blood" was confusing to me. In Ghana, samples were collected prior to donation. In Burkina Faso and Gabon, samples were collected after blood donation. Where exactly was the plasma collected after centrifugation of EDTA whole blood? And in other countries? What were the samples? Information regarding HBsAg and HBV DNA should be placed in the respective topics, as well as data analysis. What does BOTIJA mean? I couldn't find it even in reference 22.

Lines 66-67: genes should be written in italics.

Study design (item 2.6, line 77) is in the wrong position. It should be the first topic of material and methods. It's still confusing to me. What are the real objectives of the study? Why include unselected children? Isn't the target audience only blood donors and chronic HBV infection?

Was the research carried out based on scientific articles? If yes, which search platforms are used? How many papers were evaluated? What are the inclusion and exclusion criteria?

In Results
Item 3.1: The results shown refer to which study? In line 57 (methods) it says only how the collection was made. The n is too small. What statistical method was used to demonstrate significance?

In item 3.2, the authors could separate the information according to the genotype in China, considering that there are two genotypes (B and C). Does it make any difference if you separate?

Why does table 2 provide information from Taiwan, Thailand, Italy, Poland and Spain? Are these countries not part of the study.

in the discussion
Authors should substantiate well the information "The superiority of female over male's immune system has been demonstrated in 192 many situations over several decades" with citation of relevant papers.

What are the limitations of the study?

What is the importance of this study in relation to the establishment of public policies regarding blood donation in the places covered by the study?

In conclusions
The authors cannot conclude anything regarding HCV as it was not the objective of the study. If the information that female donors are safer than male and female is correct, what are the implications for policies involving blood donations worldwide?

Author Response

Answers to Reviewer 2

Title
The authors should be more specific in citing the location (continent) of areas of high HBV prevalence, as they were only China, Iran, Lebanon, Tunisia, Ghana, Burkina Faso and Gabon.

Reply. In order to keep the title short, Asian and African high or medium prevalence areas was modified.

Abstract
Abstract is confused. The study's objectives are unclear. Authors should decrease the amount of results (such as p-values, for example). It has a lot of information that fits better in the introduction.
The title does not match the information in the abstract. The authors describe many serological markers, such as anti-HBs+, that are unrelated to chronic hepatitis B. How important is information regarding Ghanaian children less than 5 years in the context of blood donors? What is the importance for the manuscript of information regarding the vaccinated in the context of chronic infection? What is the methodology used in the study? Include statistical analysis.

Reply. We clarified that HBV markers were studied in blood donors. Data are important in an abstract and we studied HBV markers, not chronic hepatitis markers. The data on viral load in children is critical because it is new data. Vaccination is becoming more and more important in young blood donors as it modulate the epidemiology of the infection; in addition it is an invaluable marker of immune response. HBsAg was screened by either ELISA or rapid tests, anti-HBc and anti-HBs by ELISA, HBV DNA load by a standardized method across sites.

Introduction
The introduction is very brief. The authors better situate the reader in relation to the choice of countries that are part of the study. For example, what determines that an area has a high, medium, or low prevalence of HBV infection? I emphasize here that in the abstract and in the title, the authors mention that they will carry out the study only in areas of high prevalence and in the objectives at the end of the introduction they also refer to areas of medium prevalence. Other points that could be included in the introduction concern: occult hepatitis B, meaning of serological markers, definition of chronic HBV infection and information regarding blood donors and HBV genotypes.

Reply. The reviewer rightly points out the discrepancy between introduction and title regarding HBV infection distribution in medium and high prevalence. This was corrected by adding ‘medium’ prevalence to the title. We believe important to keep the introduction short and straight to the topic which is immune control of HBV according to sex. Responding to reviewer 1 comments already enlarged the introduction.

Line 38: the authors place several studies and present only one reference. You should have more.

Reply. Two references were added: Voysey M, Barker CI, Snape MD, Kelly DF, Trück J, Pollard AJ. Sex-dependent immune responses to infant vaccination: an individual participant data meta-analysis of antibody and memory B cells. Vaccine. 2016;34:1657-64. And Klein SL, Marriott I, Fish EN. Sex-based differences in immune function and responses to vaccination. Trans R Soc Trop Med Hyg. 2015;109:9-15.

The objectives are still not very clear. Will the study period be 15 (abstract) or 20 years (introduction)?

Reply. This has been modified to 15 years instead of two decades in introduction.

Line 49: The sentence "Samples included in the study were collected either as serum such as pre-donation screening in Ghana or post-donation screening in Burkina Faso or Gabon or plasma collected after centrifugation of EDTA whole blood" was confusing to me. In Ghana, samples were collected prior to donation. In Burkina Faso and Gabon, samples were collected after blood donation. Where exactly was the plasma collected after centrifugation of EDTA whole blood? And in other countries? What were the samples? Information regarding HBsAg and HBV DNA should be placed in the respective topics, as well as data analysis. What does BOTIJA mean? I couldn't find it even in reference 22.

Reply. Samples collected in centrifugated plasma ‘in other countries’.

Full name of BOTIA acronym was added.

Lines 66-67: genes should be written in italics.

Reply: This has been modified as requested.

Study design (item 2.6, line 77) is in the wrong position. It should be the first topic of material and methods. It's still confusing to me. What are the real objectives of the study? Why include unselected children? Isn't the target audience only blood donors and chronic HBV infection?

Reply. Objective of the study is clearly indicated in the first sentence of study design: examine the assumption that HBV markers …in children or adults differed between male and females. Infection mostly occurs in young age justifying children data and persist in adults such as blood donors. Adequate data was available in both situations making the study complete.

Was the research carried out based on scientific articles? If yes, which search platforms are used? How many papers were evaluated? What are the inclusion and exclusion criteria?

Reply. The study was not based on reviewing scientific articles but selecting appropriate data from published studies examined through a different angle.

In Results

Item 3.1: The results shown refer to which study? In line 57 (methods) it says only how the collection was made. The n is too small. What statistical method was used to demonstrate significance?

Reply. Admittedly the numbers are small, but the difference is significant using the methods indicated in section 2.7.

In item 3.2, the authors could separate the information according to the genotype in China, considering that there are two genotypes (B and C). Does it make any difference if you separate?

Reply. In South China, genotype B is dominant (approximately 65%) but the genotype of each HBsAg positive was not determined making genotype-specific analysis not feasible.

Why does table 2 provide information from Taiwan, Thailand, Italy, Poland and Spain? Are these countries not part of the study.

Reply. As indicated in Table 1, references from which the data was collected are indicated. They are intended to enlarge the points pertaining to the study itself.

In the discussion

Authors should substantiate well the information "The superiority of female over male's immune system has been demonstrated in many situations over several decades" with citation of relevant papers.

Reply. References supporting this statement have been added: 1-6.

What are the limitations of the study?

Reply. As requested by reviewer 1, limitations of the study have been added: “It is a limitation of the study that in none of the countries where data was collected, the number of screened samples and the prevalence of HBsAg did not allow to reach statistical significance. However, the trends observed were highly reproducible between countries and HBV genotypes.”

What is the importance of this study in relation to the establishment of public policies regarding blood donation in the places covered by the study?

Reply. As indicated in the reply to the similar question from reviewer 1, in most developing countries and some developed countries, female blood donors are few. Emphasizing their safety advantage might be an argument to promote female blood donation.

In conclusions

The authors cannot conclude anything regarding HCV as it was not the objective of the study. If the information that female donors are safer than male and female is correct, what are the implications for policies involving blood donations worldwide?

Reply. The conclusion regarding HCV is supported by a reference and it adds to the main point encouraging female blood donation. This, as well as the last sentence of conclusion, places the HBV data into a larger context.

Round 2

Reviewer 2 Report

The authors considered all suggestions and greatly improved the manuscript.